# Liquid Biopsy-Derived DNA Sources as Tools for Comprehensive Mutation Profiling in Multiple Myeloma: A Comparative Study

**DOI:** 10.3390/cancers14194901

**Published:** 2022-10-07

**Authors:** Robbe Heestermans, Wouter De Brouwer, Ken Maes, Isabelle Vande Broek, Freya Vaeyens, Catharina Olsen, Ben Caljon, Ann De Becker, Marleen Bakkus, Rik Schots, Ivan Van Riet

**Affiliations:** 1Department of Clinical Biology, Vrije Universiteit Brussel (VUB), Universitair Ziekenhuis Brussel (UZ Brussel), Laarbeeklaan 101, 1090 Brussels, Belgium; 2Department of Hematology, Vrije Universiteit Brussel (VUB), Universitair Ziekenhuis Brussel (UZ Brussel), Laarbeeklaan 101, 1090 Brussels, Belgium; 3Research Group Hematology and Immunology, Vrije Universiteit Brussel (VUB), Laarbeeklaan 103, 1090 Brussels, Belgium; 4Clinical Sciences, Research Group Reproduction and Genetics, Centre for Medical Genetics, Vrije Universiteit Brussel (VUB), Universitair Ziekenhuis Brussel (UZ Brussel), Laarbeeklaan 101, 1090 Brussels, Belgium; 5Department of Oncology and Hematology, VITAZ, Moerlandstraat 1, 9100 Sint-Niklaas, Belgium; 6Brussels Interuniversity Genomics High Throughput Core (BRIGHTcore), Vrije Universiteit Brussel (VUB), Université Libre de Bruxelles (ULB), Laarbeeklaan 101, 1090 Brussels, Belgium

**Keywords:** multiple myeloma, cell-free DNA, liquid biopsy, mutation profiling, extracellular vesicle, circulating tumor cells

## Abstract

**Simple Summary:**

Multiple myeloma (MM) is characterized by an expansion of plasma cells in the bone marrow (BM). The genetics of MM are highly complex with multiple mutations and genetic subpopulations of tumor cells that arise during the disease evolution, affecting prognosis and treatment response. Standard bone marrow DNA analysis requires an invasive sample collection and does not always reflect the complete mutation profile. Therefore, we examined the possibility to use peripheral blood-based liquid biopsies as an alternative DNA source for mutation profiling. By comparing DNA from circulating tumor cells with circulating tumor-derived vesicles and cell-free DNA (cfDNA), we found that the latter provided the best concordance with bone marrow DNA and also showed mutations derived from myeloma cell populations that were undetectable in bone marrow. Our comparative study indicates that cfDNA is the preferable circulating biomarker for genetic characterization in MM and can provide additional information compared to standard BM analysis.

**Abstract:**

The analysis of bone marrow (BM) samples in multiple myeloma (MM) patients can lead to the underestimation of the genetic heterogeneity within the tumor. Blood-derived liquid biopsies may provide a more comprehensive approach to genetic characterization. However, no thorough comparison between the currently available circulating biomarkers as tools for mutation profiling in MM has been published yet and the use of extracellular vesicle-derived DNA for this purpose in MM has not yet been investigated. Therefore, we collected BM aspirates and blood samples in 30 patients with active MM to isolate five different DNA types, i.e., cfDNA, EV-DNA, BM-DNA and DNA isolated from peripheral blood mononucleated cells (PBMNCs-DNA) and circulating tumor cells (CTC-DNA). DNA was analyzed for genetic variants with targeted gene sequencing using a 165-gene panel. After data filtering, 87 somatic and 39 germline variants were detected among the 149 DNA samples used for sequencing. cfDNA showed the highest concordance with the mutation profile observed in BM-DNA and outperformed EV-DNA, CTC-DNA and PBMNCs-DNA. Of note, 16% of all the somatic variants were only detectable in circulating biomarkers. Based on our analysis, cfDNA is the preferable circulating biomarker for genetic characterization in MM and its combined use with BM-DNA allows for comprehensive mutation profiling in MM.

## 1. Introduction

During the last decade, there has been an impressive progress in the treatment modalities of multiple myeloma (MM). Despite the availability of new drugs and innovative therapeutic options, most patients will still relapse and become refractory to treatment at some point in the disease evolution [1]. MM is characterized by a patchy tumor infiltration pattern in the bone marrow (BM), implying that the analysis of a single bone marrow sample during a routine clinical diagnostic or follow-up procedure is likely to underestimate the true heterogeneity within the tumor. Several landmark studies have demonstrated (spatial) genetic heterogeneity in the bone marrow of MM patients [2,3,4,5,6]. Moreover, an increase in mutational complexity occurs throughout MM disease progression, affecting prognosis and response to therapy, which necessitates a flexible tool for genetic profiling [4,7,8,9,10,11,12,13]. Therefore, there is an increasing interest in the development of peripheral blood-based monitoring methods. This would also allow an easier and non-invasive sampling procedure, facilitating a more frequent disease follow up. A limited number of studies have (individually) compared *cell free DNA* (cfDNA) and DNA derived from *circulating tumor cells* (CTCs) with BM-DNA and reported good concordance of the observed mutation profile [3,14,15,16,17,18,19]. In addition, circulating biomarkers can reveal unique genetic variants that are not detected in bone marrow [3,10,14,15,20,21]. However, all but one of these studies only investigated the concordance between BM-DNA and one other matched liquid biopsy-derived DNA source.

*Extracellular Vesicles* (EVs) recently emerged as a promising new biomarker to study tumor-related genetic alterations [22,23,24]. EVs and exosomes carry proteins, lipids, various types of RNA and double-stranded DNA [22,23,24,25]. Since EVs are released by tumor cells, it can be assumed that they reflect the molecular profile of these cells [26]. Hence, EVs isolated from liquid biopsies might serve as biomarker for non-invasive cancer diagnostics and disease follow up. Because EVs have an intrinsic robustness due to their lipid bilayer and can contain larger DNA fragments, long-term sample storage and sequencing applications might be facilitated when using EV-DNA [25,27,28,29]. In contrast, cfDNA consists of small DNA fragments (with mutant circulating tumor DNA (ctDNA) even more fragmentated than normal cfDNA) and tends to be more unstable than EV-DNA when collected in routinely used EDTA tubes [30,31]. It therefore often necessitates prompt sample processing and/or the use of expensive collection tubes designed for cfDNA isolation, and its fragmented nature might hamper downstream sequencing applications. Studies have successfully detected *KRAS* and *TP53* variants in EV-DNA from patients with pancreatic cancer [27,32,33,34]. Likewise, several studies demonstrated the detection of the therapeutically important *EGFR* mutations in the EV-DNA of NSCLC patients [35,36]. In a recent study in children with acute myeloid leukemia, the authors observed a 90% concordance between leukemia-associated mutations found in the BM and EV-DNA [37]. However, to the best of our knowledge, this emerging biomarker has not been studied yet for the purpose of mutation profiling in MM. Moreover, a thorough comparative study between the forementioned circulating biomarkers (cfDNA, CTCs, peripheral blood mononucleated cells (PBMNCs)-DNA and EV-DNA) as non-invasive tools for comprehensive genetic tumor characterization is currently lacking. Therefore, our study aims to provide this missing evidence by comparing the mutation profile obtained from five different DNA sources, isolated from matched BM and blood samples in patients with active MM.

## 2. Materials and Methods

### 2.1. Patient Samples Collection and Pre-Analytical Sample Processing

Matched bone marrow aspirates and blood samples were collected in EDTA tubes in 30 multiple myeloma patients with active disease. All included patients signed an informed consent form prior to sample collection and this study was approved by the Ethical Committees of both UZ Brussel and VITAZ (B.U.N. 14321733078). Figure 1 provides an overview of the sample processing workflow. Blood samples (30 mL) were centrifuged to separate plasma from the whole blood. The EVs were isolated from a maximum of 4 mL of plasma according to the recommendations of the exoEasy Maxi kit (Qiagen, Venlo, The Netherlands) and were eluted in a volume of 400 µL [38]. EV-DNA was extracted with the QIAamp DNA Micro kit (Qiagen), which has been used for this purpose previously [32,37]. Instead of 100 µL of whole blood, the 400 µL of eluate from the exoEasy maxi kit (Qiagen) was used as the starting material. As a consequence, the volume of the added reagents was quadruplicated to 80 µL of ATL buffer (Qiagen), 40 µL of Protease (Qiagen), 400 µL of AL buffer (Qiagen) and 200 µL of ethanol (>99.8%). The mixture was applied in two steps onto the QIAamp MinElute column (Qiagen) and DNA was eluted in 50 µL. cfDNA was isolated with the QIAamp Circulating Nucleic Acid kit (Qiagen) and eluted in 60 µL. To enrich the mononucleated cells (MNCs) from whole blood and bone marrow samples, a ficoll density gradient centrifugation was used. A fraction of 1 × 10^6^ cells from the PBMNCs was used as an unenriched fraction whereas the remainder of the PBMNCs was used for the enrichment of CTCs with CD138-coated beads on the immunomagnetic separation device autoMACS^®^ (Miltenyi Biotec, Leiden, The Netherlands). Plasmacytosis in the BM samples was flowcytometrically determined on the FACSLyric^TM^ (BD, Franklin Lakes, NJ, USA) using antisera against CD38 (BD), CD138 (IQ Products, Groningen, The Netherlands) and CD45 (Agilent Technologies Inc., Santa Clara, CA, USA). DNA was extracted from PBMNCs, BM MNCs and CTCs using the QIAamp DNA Blood Mini kit (Qiagen). BM-DNA and PBMNCs-DNA was eluted in 100 µL and CTC-DNA was eluted in 80 µL. DNA samples were quantified with the Qubit^TM^ dsDNA BR Assay Kit (Thermo Fisher Scientific, Waltham, MA, USA) using the VICTOR Nivo^TM^ plate reader (PerkinElmer, Waltham, MA, USA) and stored at −20 °C until library preparation.

### 2.2. Culturing of Human Myeloma Cell Lines (HMCLs)

As positive controls, we selected several Human Myeloma Cell Lines (HMCLs) (OPM-2, U266 and RPMI-8226) that are known to harbor previously described genetic variants [39,40,41,42]. OPM-2 and RPMI-8226 were purchased from ATCC (Manassas, VA, USA) whereas U266 was kindly provided by Dr. Kenneth Nilsson (Uppsala University, Sweden). The cells were cultured according to standard conditions in a humidified incubator at 37 °C with 5% CO_2_ using an RPMI culture medium (Gibco, Waltham, USA) with 10% Fetal Bovine Serum (FBS) (Gibco), 1% L-glutamine (Gibco) and 1% Penicillin/streptomycin (Gibco). After harvesting, the DNA was extracted out of the cultured cells according to the recommendations of the QIAamp DNA Blood Mini kit (Qiagen). The DNA was eluted into 200 µL of elution buffer (Qiagen) and was stored at −20 °C until library preparation.

### 2.3. Library Preparation and Targeted Gene Sequencing

Five different matched DNA types (i.e., BM-DNA, PBMNCs-DNA, EV-DNA, CTC-DNA and cfDNA) were used for targeted gene sequencing with a panel of 165 genes that are associated with solid and hematological malignancies. This panel contained several genes that are frequently mutated in MM and/or are of prognostic and therapeutic interest. Sequencing was performed in collaboration with BRIGHTcore (UZ Brussel, Vrije Universiteit BrusselUniversité Libre de Bruxelles (Brussels, Belgium)). A complete list of the targeted genes can be found in Appendix A. For EV-DNA and cfDNA, up to 25 µL of DNA-containing eluate was used in the KAPA Hyper Prep (Roche Sequencing, Pleasanton, CA, USA) library preparation according to the manufacturer’s recommendations with several modifications. The volumes were reduced with a factor 2. Unique Dual Indexed (UDI) adapters with a 7-bases P7 Unique Molecular Index (UMI) designed by BRIGHTcore (Integrated DNA Technologies, Coralville, IA, USA) were used at a concentration of 1.5 μM. A 1x post ligation bead cleanup with AMPure XP beads (Beckman Coulter Life Sciences, Indianapolis, IN, USA) was performed and a total of 15 PCR cycles were applied to obtain a sufficient library for target enrichment. For high quality/quantity DNA obtained from peripheral blood and bone marrow samples and HMCLs, the libraries were constructed on 150 ng of input DNA with the KAPA HyperPlus kit (Roche Sequencing) according to the manufacturer’s recommendations, with three modifications: (1) an enzymatic fragmentation for 20 min at 37 °C was used to obtain DNA insert sizes of on average 200 bp, (2) the usage of 15 µM of our in-house designed UDI/UMI adapters and (3) a total of 7 PCR cycles were applied to obtain a sufficient library for target enrichment. Target enrichment was performed according to version 5.0 of the manufacturer’s instructions with a homebrew Roche SeqCap EZ Choice probemix (Roche Sequencing). Pre-capture pooling was limited to a maximum of 8 samples for a total of 1 µg of pooled library. In contrast to the instructions, the xGen Universal Blockers TS Mix (Integrated DNA Technologies) replaced the sequence-specific blocking oligos and the final PCR was limited to 12 PCR cycles. The final libraries were qualified on the AATI Fragment Analyzer (Agilent Technologies Inc.) using the DNF-474 High-Sensitivity NGS Fragment Analysis Kit (Agilent Technologies Inc.) and quantified on the Qubit 2.0 with the Qubit dsDNA HS Assay Kit (Thermo Fisher Scientific). Per sample, a minimum of 14.5 million 2x100 bp reads were generated on the Illumina NovaSeq 6000 system (Illumina Inc., San Diego, CA, USA), with the NovaSeq 6000 S2 Reagent Kit (200 cycles). For this, 1.9 nM libraries were denatured according to manufacturer’s instruction.

### 2.4. Sequencing Data Processing, Variant Filtering and Interpretation

Illumina’s bcl2fastq algorithm (version 2.19) was used to convert the raw basecall files into fastq files, after which the reads were aligned to the human reference genome (hg19) using BWA (version 0.7.10-r789). Picard (version 1.97) was used to mark the duplicate reads. Genome Analysis Toolkit (GATK) (version 3.3) was used to provide post-processing of the aligned reads, which consisted of realignment around insertions/deletions (indels) and base quality score recalibration. The quality control on the post-processed aligned reads was performed by using samtools flagstat (version 0.1–19) and picard HsMetrics (version 1.97), which were also used to investigate the total number of reads, the percentage of duplicate reads, the mean coverage on targets and the percentage of on-target, near-target and off-target bases. Variants were called with GATK Mutect2 (version 4.0.12.0) in tumor-only mode and annotation was conducted with Annovar and Alamut batch version 1.11 and Alamut database version alamut_db-1.5-2021.06.01.db. These tools provide information for each variant about its location in a gene/protein, its occurrence in multiple normal populations and its (predicted) effect at the protein level. The filtering and interpretation of the variants was performed based on the recommendations of the ComPerMed guidelines (version 3, April 2021) using an in-house-designed R script [43]. This script filters variants at a 1% population frequency based on esp6500, 1000 g and Gnomad annotation by Annovar (annovar_2018Apr16). The following criteria were used to filter the called variants: variants known as artefacts/recurrent variants were excluded; synonymous variants were excluded; variants with a MAF > 0.1% were excluded; and variants with a VAF < 3% in all matched DNA samples of a patient were excluded. A minimum of 25 mutant reads and a VAF of ≥ 3% in at least one of the five matched DNA samples was required to include a variant for further analysis and only the called variants were reported. As a final control, after data filtering, the called variants were subjected to manual inspection of the aligned reads in IGV (version 2.6.3) to confirm their presence [44].

### 2.5. Monoclonal Immunoglobulin Detection with Next-Generation Sequencing

For patients in whom discrepancies were observed between the mutation profile of BM-DNA and circulating biomarker DNA, we evaluated the percentage of monoclonal immunoglobulin sequences within the BM-DNA samples to confirm the presence of tumor DNA within these samples. An NGS-based approach was used targeting the *IgH* and *IgK* locus as was previously described by our research group [45].

### 2.6. Statistics

A Kruskal–Wallis test and Mann–Whitney U test were used for comparisons between the independent variables and a Wilcoxon signed-rank test for comparisons between the paired samples. Pearson correlation and Spearman rank correlation coefficients were calculated to assess the correlation between the continuous and discrete variables, respectively. A *p*-value < 0.05 was considered to be statistically significant. Statistical analyses were performed using Medcalc (version 14.8.1, Ostend, Belgium).

## 3. Results

### 3.1. Patient Characteristics and Obtained DNA Yields

Blood and bone marrow samples were obtained from three groups of ten patients (10 at diagnosis, 10 at first relapse and 10 at second or later relapse). The mean age at sample collection was 73 years (range 64–88 years) and our study group consisted of 57% (17/30) males and 43% (13/30) females. Additional patient characteristics can be found in Table 1 and Appendix A. In all of the 30 included patients, five different DNA types were isolated, except for one patient where no CTC enrichment was performed due to low PBMNCs counts. Thus, a total of 149 DNA samples were used for targeted gene sequencing. The absolute DNA yields are shown in Figure 2 (Box and Whiskers plots, logarithmic scale). The median DNA yield derived from the BM samples was 6655 ng (range 1720–46,200 ng). The PBMNCs samples had a median DNA yield of 5890 ng (range 1740–39,700 ng). The median cfDNA and EV-DNA yields were 138 ng (range 18–13,386 ng) and 10 ng (range 5–275 ng), respectively. CTC-DNA showed a median DNA yield of 315 ng (range 48–1888 ng). A significant correlation was found between the LDH concentration measured at the sample draw and the yields of BM-DNA (r = 0.64, *p* < 0.001), EV-DNA (r = 0.40, *p* < 0.05) and PBMNCs-DNA (r = 0.70, *p* < 0.001). No significant differences in the DNA yields were observed between the different MM patient groups (Kruskal–Wallis test).

### 3.2. Detection of Previously Described Genetic Variants in HMCLs

Table 2 provides an overview of the variants identified in the HMCLs. In OPM-2, we successfully detected the previously described *DIS3* p.Tyr121Ser and *FAM46C* p.Glu178Ala variants [39,40]. For both variants, we observed a VAF of 100%, indicating that these were homozygous variants. In U266, we detected the homozygous variant *TP53* p.Ala161Thr, previously described by Moreaux et al. (2011) [42]. We also identified the *BRAF* p.Lys601Asn variant: the observed VAF of +/−65% corresponds with the VAF Lionetti et al. (2015) described [41]. Finally, *KRAS* p.Gly12Ala was detected in RPMI-8226 [42]. 

### 3.3. Genetic Pathway Involvement and Biological Significance of Detected Variants in MM Patients

After the filtering of raw sequencing data, a total number of 126 variants was detected with the 165-gene panel (Appendix A). This was further divided into 87 somatic variants and 39 variants that are most likely of germline origin (VAFs of +/−50% in all paired DNA samples). The somatic variants consisted of 65 non-synonymous SNVs, four splice site variants, five stopgains, one non-frameshift deletion and twelve frameshift deletions/insertions. Among the germline variants were 35 non-synonymous SNVs, two frameshift deletions, one splice site variant and one non-frameshift insertion. At least one somatic variant was observed in 87% (26/30) of the included patients. The distribution of the somatic variants across the 165-gene panel together with their biological significance is shown in Figure 3A. All somatic variants were located within 41 of the 165 targeted genes (25%). The three commonly mutated genes *NRAS*, *KRAS* and *TP53* harbored together over 25% (22/87) of the detected somatic variants and 50% (15/30) of the included patients had at least one somatic variant in any of these three genes. Among the most frequently mutated genes were also *DNMT3A* (17%) and *FAM46C* (13%). Of note, all of the four patients who had the *FAM46C* somatic variants had a hyperdiploid karyotype and both patients with the *CCND1* somatic variants had a t(11;14). Figure 3B shows the genetic pathways (of which the majority have an important role in MM pathogenesis) that are affected by genetic variants. When considering the somatic variants (blue), the pathways involved in cell proliferation, apoptosis and survival were most frequently affected. Considerable amounts of somatic variants were found in genes of the PI3K/Akt pathway (34%, 30/87), the MAPK pathway (29%, 25/87) and the mTOR signaling pathway (26%, 23/87). The genes involved in p53 signaling (17%, 15/87) and the cell cycle pathway (16%, 14/87) were also affected by somatic variants to a considerable degree. Germline variants (orange) mainly affected the MAPK pathway (21%, 8/39) and the pathways involved in DNA repair (21%, 8/39) and PI3K/Akt signaling (18%, 7/39) (of note, some genes act in multiple pathways; principal source for pathway analysis: KEGG database) [46,47,48]. The majority of the observed somatic variants were classified as VUS (54%, 47/87) while 26% (23/87) and 20% (17/87) were classified as pathogenic and likely pathogenic, respectively (Figure 3A). Of note, 63% (19/30) of the patients had at least one pathogenic or likely pathogenic somatic variant. Almost all the germline variants were classified as VUS (95%, 37/39), while only two LoF variants (5%) in *PMS2* and *CDKN2B* were classified as likely pathogenic. 

### 3.4. Differential Mutational Burden across MM Disease Stages and Correlation with Clinical Outcome

When comparing the mutational burden across the different MM disease stages, we observed a median number of one (range 0–7) somatic variant in patients with new diagnosis (*n* = 10), whereas patients with first relapse (*n* = 10) and second or later relapse (*n* = 10) had a median number of one (range 0–5) and five (range 3–7) somatic variants, respectively. There was a significant difference in the number of somatic variants between the three MM disease stages (*p* = 0.003, Kruskal–Wallis test). More specifically, patients with second or later relapse had significantly more somatic variants compared to patients with first relapse and new diagnosis (*p* = 0.001 and *p* = 0.01, respectively, Mann–Whitney U test). Consistent with this result, a significant correlation was found between the somatic variant count and the number of treatment lines patients had received at the time of sample collection (rho = 0.49, *p* = 0.006). No significant difference in somatic variant count was observed between patients with new diagnosis and first relapse (*p* = 0.73, Mann–Whitney U test). When correlating the mutational burden with clinical outcome variables, we found that the number of somatic variants was significantly higher in patients who had died (*n* = 16) before the conclusion of this study compared to patients who were still alive (*n* = 14) (median number 4.5 versus 1, *p* = 0.02, Mann–Whitney U test). In parallel, the number of (likely) pathogenic somatic variants was significantly higher among deceased patients (*p* = 0.002, Mann–Whitney U test) and in patients with second or later relapse compared to earlier disease stages (*p* = 0.01, Mann–Whitney U test).

### 3.5. Detectability of Genetic Variants by Circulating Biomarkers and Concordance with Bone Marrow DNA

Sequencing results were obtained from 148 DNA samples, as library preparation failed for one EV-DNA sample (2018-011) and a CTC-DNA sample was not available in one patient (2018-013). The mean target coverage was 2220x (range 68x–4147x) for BM-DNA, 2344x (range 255x–4138x) for PBMNCs-DNA, 1260x (range 257x–2172x) for cfDNA, 870x (range 299x–1538x) for CTC-DNA and 633x (range 12x–2736x) for EV-DNA samples. Detailed results including the VAFs and coverage observed in each matched DNA sample for the detected variants are provided in Appendix A. The variants that were classified as germline variants (*n* = 39) were detectable in all of the five matched DNA samples. Exceptions to this are the fore-mentioned missing EV-DNA and CTC-DNA sample and the EV-DNA sample of patient 2018-009 where the coverage was very low (four reads) and one germline variant was not detected.

Figure 4A summarizes the detectability of somatic variants in the five different DNA types that we studied. We excluded the somatic variants detected in patient 2018-013 (*n* = 3) to calculate the detection rate in CTC-DNA and the somatic variants detected in patient 2018-011 (*n* = 1) to calculate the detection rate in EV-DNA. Overall, 77% (67/87) of the somatic variants were detected in BM-DNA and at least one blood-derived DNA source. In primary tumor BM-DNA, we detected 84% (73/87) of all the reported somatic variants. However, cfDNA permitted the detection of 87% (76/87) of the somatic variants, compared to 80% (67/84), 74% (64/86) and 49% (43/87) in CTC-DNA, EV-DNA and PBMNCs-DNA, respectively. A statistical analysis showed that the number of somatic variants detected in cfDNA was significantly higher compared to EV-DNA (*p* = 0.002) and PBMNCs-DNA (*p* < 0.001), but not compared to CTC-DNA (*p* = 0.16) (Wilcoxon signed-rank test). Of note, BM-DNA and cfDNA together permitted us to detect 99% (86/87) of all the reported somatic variants. When calculating the concordance with the somatic variants observed in BM-DNA (*n* = 73), 86% (63/73) were also detected in matched cfDNA samples compared to 83% (58/70) in CTC-DNA, 73% (53/73) in EV-DNA and 59% (43/73) in PBMNCs-DNA. We were able to detect significantly more of the BM positive somatic variants in cfDNA compared to EV-DNA and PBMNCs-DNA (both *p* < 0.01, Wilcoxon signed-rank test). No other significant differences between the circulating biomarkers were observed. When correlating the VAFs of the five different DNA types we studied, we found an almost perfect linear relationship between the VAFs observed in cfDNA and EV-DNA (Figure 4B) (r = 0.9538, *p* < 0.0001).

### 3.6. Discordant Results between Blood and Bone Marrow Compartment and Correlation with Monoclonal Immunoglobulin Percentages and Plasmacytosis

Although for most variants a high degree of concordance was observed between the mutation profile observed in the BM-DNA and circulating biomarkers, there were also some discrepant cases that required closer inspection. Table 3 shows the somatic variants that were selectively detected in BM-DNA. This comprised 7% (6/87) of all the detected somatic variants and involved five patients (one new diagnosis and four at first relapse). The DNA yields measured in these five patients were below the median in 4/5 cfDNA, 1/5 EV-DNA, 4/5 PBMNCs-DNA and 3/5 CTC-DNA samples. Related to this, the coverage at the nucleotide position of the particular variants was below 500 reads in the EV-DNA and CTC-DNA samples of three patients. The observed VAFs in BM-DNA were moderately low and ranged from 3.2% to 6.4%. Of note, patient 2020-024, who carried a BM-only likely pathogenic *RB1* variant, also had four somatic variants that were only detected in circulating biomarkers, which are listed in Table 4. In total, 16% (14/87) of the somatic variants were only detected in circulating biomarkers and not in BM-DNA, including five pathogenic variants and two likely pathogenic variants. Of note, cfDNA permitted us to detect 93% (13/14) of these BM negative somatic variants compared to 85% (11/13), 64% (9/14) and 0% in EV-DNA, CTC-DNA and PBMNCs-DNA, respectively. In two patients (i.e., 2018-005 and 2019-007), extramedullary (EM) disease was strongly suspected and this was later confirmed in patient 2019-007. The percentage of the BM plasma cells measured with flow cytometry and the percentage of the monoclonal Ig sequence reads in the BM are shown in Table 3 and 4 for the discrepant cases if available. In 5/8 patients with BM negative somatic variants, these data showed the presence of a considerable percentage of monoclonal Ig sequence reads in the BM-DNA sample, confirming the presence of tumor DNA within these BM-DNA samples. In one patient in whom EM disease was suspected, the percentage of monoclonal Ig was low (0.7%). For two patients, no data were available. Although carrying BM negative somatic variants, 5/8 patients also had BM positive somatic variants (Appendix A), indicating the presence of tumor DNA within these BM-DNA samples as well.

## 4. Discussion

To the best of our knowledge, our study is the first to report comparative mutation profiling data obtained with BM-DNA and the simultaneous analysis of all currently available liquid biopsy-derived DNA sources in MM. Moreover, we are the first to report targeted sequencing results obtained in the EV-DNA of MM patients. The sequencing method and bioinformatics pipeline we used was validated for reporting the variants detected in cellular DNA in a clinical setting. In this context, stringent filtering criteria were required to ensure the correct interpretation of the sequencing data and to avoid reporting false positive results. Therefore, the variants were filtered and interpreted based on the ComPerMed guidelines, which are consensus guidelines involving experts in the field of genetic diagnostics across Belgium [43]. As such, only variants that met pre-set laboratory thresholds for coverage and VAF were reported. Every variant we reported met this criterium for coverage and VAF in at least one of the five matched DNA samples, ensuring the validity of the reported variants.

In this comparative study, a good overall concordance between BM and liquid biopsy-derived DNA was found. We detected 77% (67/87) of all the reported somatic variants in BM-DNA and at least one circulating biomarker. As was shown in Figure 4A, cfDNA yielded the highest overall somatic variant detection rate (87%) of all the five DNA types we studied. Similarly, cfDNA had the highest concordance with the mutation profile observed in primary tumor BM-DNA. In both comparisons, cfDNA significantly outperformed EV-DNA and PBMNCs-DNA. Although PBMNCs-DNA has previously been used as constitutional control DNA, we also detected somatic variants in this DNA source [4]. The number of somatic variants we detected in PBMNCs-DNA was, however, significantly lower compared to the other circulating biomarkers we investigated. This contrasts with the high somatic variant detection rate in PBMNCs-DNA reported by Coffey et al. (2019), where ultradeep sequencing was used [49]. Given the considerable costs associated with using ultradeep sequencing and the inferior performance of PBMNCs-DNA that we observed, we considered PBMNCs-DNA a less-suitable tool for non-invasive mutation profiling in MM.

Although EV-DNA and exosome DNA were found to be superior to cfDNA in detecting tumor-associated mutations in some earlier studies in patients with solid malignancies, this was not confirmed in our study in MM patients [33,34,35,36,50]. We used the commercial exoEasy Maxi kit (Qiagen) for EVs purification. Previous research has shown that EVs purified with this kit are of equal quality as those purified with ultracentrifugation-based methods, while the latter method is very laborious which hampers its use in a clinical laboratory setting [28,38]. It was particularly challenging to obtain sufficient EV-DNA for targeted sequencing, although the measured concentrations were similar to those of some previous studies on EV-DNA [32]. The low EV-DNA yields that we obtained (median 10 ng) can partly explain the lower somatic variant detection rate in EV-DNA compared to cfDNA, as for 14/22 (64%) of the somatic variants that we could not detect in EV-DNA coverage was below 500 reads (Appendix A). The median DNA yield in cfDNA we obtained was 14 times higher compared to EV-DNA. Hence, increasing coverage by increasing the EV-DNA yield up to the level of cfDNA would require a 14-fold increase in sample volume (over 100 mL of blood) when using the methodology we described. Such an extended sample volume is less evident for clinical applications. Because of these limitations and the inferior performance of EV-DNA compared to cfDNA, other circulating biomarkers seem to be more suitable tools for non-invasive mutation profiling.

The differences in somatic variant detectability between cfDNA and CTC-DNA were less pronounced when compared to the other circulating biomarkers discussed above (Figure 4) and were not statistically significant. Both biomarkers also revealed variants that were not shared, as was also reported by Manier et al. (2018) [3]. However, more variants were uniquely detected in cfDNA and not in CTC-DNA. The added value of the immunomagnetic enrichment of the CTCs from PBMNCs was confirmed, as this permitted a detection of 67% of the somatic variants that were undetected in PBMNCs. The concordance of 83% between CTC-DNA and the mutation profile in BM-DNA we reported was equal to that observed in a recent study of Garcés et al. (2020) [19]. However, cfDNA still resulted in a higher detection rate in all subgroups of somatic variants. Moreover, the use of CTC-DNA requires a more complex sample processing procedure, involving the isolation of PBMNCs from blood samples prior to the immunomagnetic enrichment of CTCs and final DNA extraction. In contrast, the isolation of cfDNA from plasma requires very little pre-analytical sample preparation and is nowadays performed in automated systems when applied in a clinical laboratory setting. This ensures better standardization and sample traceability. When taking this into account along with our observation that a combined analysis of BM-DNA and cfDNA permitted the detection of 99% of all the reported somatic variants, we consider cfDNA to be the most suitable liquid biopsy-derived DNA source for comprehensive mutation profiling.

When focusing on the mutation profiles that we observed among the MM patients in our study, a number of interesting observations were made in relation to the genetic characterization in MM patients. In agreement with previous sequencing studies, somatic variants were most frequently found in *NRAS*, *KRAS* and *TP53*, affecting mainly the PI3K/Akt and MAPK pathway and p53 signaling [8,13,51,52,53]. We confirmed the previously reported association between hyperdiploidy and *FAM46C* somatic variants, as well as the association between *CCND1* somatic variants and t(11;14) [54]. We also observed an association between high mutational burden and/or *TP53* mutations on one hand and an inferior prognosis on the other hand, as was previously demonstrated [21,54,55,56]. The functional impact of the germline variants we detected remains unclear, as all but two were classified as VUS and most of the genes carrying germline variants were only affected in one patient. Wei et al. (2018) showed that truncating *KDM1A* germline variants confer to a nine-flod increased risk to develop MM [57]. Unfortunately, *KDM1A* was not included in the 165-gene panel we used in our study. However, we detected four germline variants in *KMT2A* and *KMT2D*, which are both epigenetic regulators like *KDM1A*. As the available evidence regarding germline variants in MM remains limited, further research is needed to better comprehend their role in MM pathogenesis [58,59]. Of note, circulating biomarkers permitted us to detect the majority of the variants discussed in this paragraph.

An interesting observation from our study was that circulating biomarkers and especially cfDNA provided an important added value by revealing additional ((likely) pathogenic) variants not detected in BM-DNA (Table 4). In 2/8 patients where this was the case, EM disease was strongly suspected (confirmed in one patient) and it is likely the BM negative somatic variants here reflect the mutation profile of the EM tumor site. As we could demonstrate the presence of myeloma-specific Ig sequence in the BM-DNA samples of 6/8 of these patients, the absence of these somatic variants in BM-DNA cannot be explained by the absence of tumor DNA in the sample. Instead, this most likely reflects the spatial genetic heterogeneity in MM as previously reported [2,6,60]. The somatic variants only detected in the blood compartment emphasize the importance of circulating biomarkers for comprehensive mutation profiling, as valuable information can be overlooked when only analyzing BM samples. However, BM-DNA inversely revealed six variants that were undetectable in all the matched circulating biomarkers of five patients (Table 3). Although not applicable to all of these patients, a majority had low circulating biomarker DNA yields with a consequently low coverage depth. Together with the relatively low VAFs observed in BM-DNA (suggesting a subclonal origin of these variants) and the low levels of CTC-DNA (suggesting low levels of circulating tumor DNA), this might explain why we failed to detect these variants. Manier et al. (2018) observed a lower tumor fraction in both cfDNA and CTC-DNA in relapsed MM patients and suggested this might be due to a lower tumor burden at early relapse [3]. As 4/5 patients with variants only detectable in BM were sampled at first relapse, this might provide an additional explanation for the observed discrepancies. When considering the discrepant results discussed in this final paragraph, it seems that a combined approach consisting of sequencing matched cfDNA and BM-DNA samples is preferable. When we analyzed both together, these two DNA sources permitted us to obtain the most complete overview of the mutation profile in our MM patients. A follow-up study using this approach to track the genetic changes throughout the MM disease course with serial samples would be highly interesting.

## 5. Conclusions

In conclusion, we have reported results from the first comprehensive comparative study between BM-DNA, cfDNA, CTC-DNA, PBMNCs-DNA and EV-DNA to characterize the mutation profile in patients with active MM. Being the first study to report targeted sequencing results obtained in EV-DNA of MM patients, we demonstrated that cfDNA outperforms all other circulating biomarkers and is the preferable biomarker to characterize the genetic tumor profile in MM. As both BM-DNA and especially cfDNA reveal unique genetic variants, a combined sequencing of matched BM-DNA and cfDNA samples is preferred when attempting to obtain a complete overview of the genetic tumor profile. The considerable amount of somatic variants restricted to the blood compartment firmly ascertains the added value of the circulating biomarkers in the diagnostic and prognostic evaluation of MM.

## Figures and Tables

**Figure 1 cancers-14-04901-f001:**
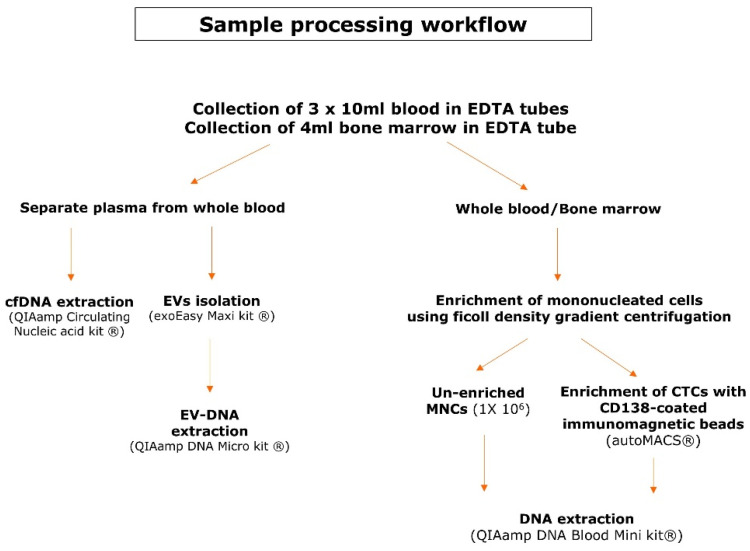
Sample processing workflow. This scheme gives an overview of the pre-analytical steps and the kits used for DNA extraction of the different DNA types.

**Figure 2 cancers-14-04901-f002:**
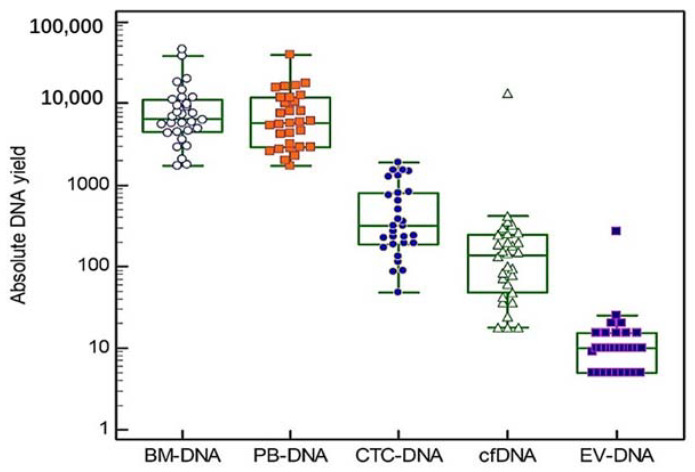
Absolute DNA yields. For each of the five DNA types we studied, individual absolute DNA yields are shown on a logarithmic scale. The central box in the boxplot indicates the values between the 25th and 75th percentile, while the line within the box represents the median. Outliers are shown as separate points above the upper limit of the boxplot.

**Figure 3 cancers-14-04901-f003:**
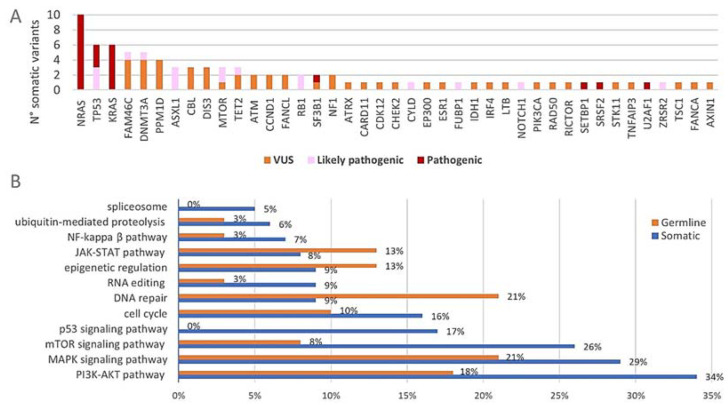
(**A**) Distribution of the somatic variants across the 165-gene panel. The absolute number of somatic variants (*n* = 87) in each of the 41 of 165 genes that carried mutations is shown. The pathogenicity of each variant is indicated with different colors. (**B**) Proportional involvement of genetic pathways by somatic and germline variants. The percentages of somatic (*n* = 87) and germline (*n* = 39) variants affecting each of the indicated pathways are shown in the graph. Some genes act in multiple pathways and are counted accordingly.

**Figure 4 cancers-14-04901-f004:**
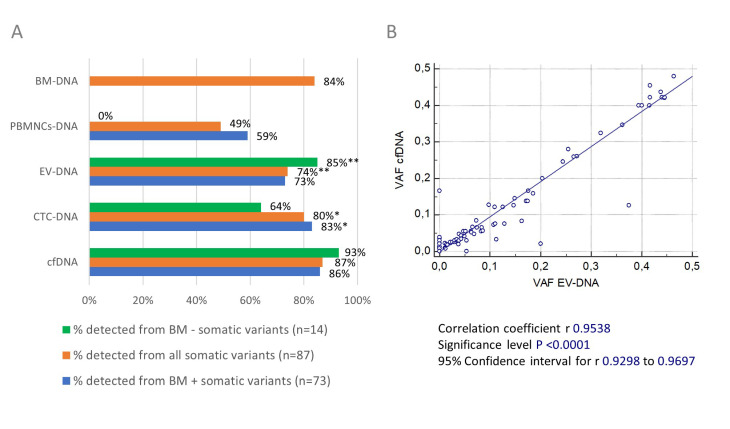
(**A**) Detectability of somatic variants in circulating biomarker and BM DNA samples. The overall detection rate of somatic variants in the five DNA types we studied is shown in orange. In blue, the concordance is shown between the somatic variants detected in BM-DNA and matched circulating biomarker DNA samples. The detection rates of somatic variants uniquely detected in the blood compartment are shown in green. * exclusion of somatic variants detected in 2018-013 (*n* = 3) to calculate % because no CTC-DNA sample was available for sequencing; ** exclusion of somatic variants detected in 2018-011 (*n* = 1) to calculate % because library preparation failed for EV-DNA sample. (**B**) Correlation between VAFs of somatic variants detected in EV-DNA and cfDNA. The scatter plot shows an almost perfect linear correlation (*p* < 0.0001) between the VAFs of somatic variants detected in EV-DNA and cfDNA. Pearson’s correlation coefficient and 95% CI were calculated.

**Table 1 cancers-14-04901-t001:** Patient characteristics according to disease stage.

	New Diagnosis (*n* = 10)	First Relapse (*n* = 10)	Second or Later Relapse (*n* = 10)
**Median age (range)**	72 (64–88)	74 (68–83)	73 (67–81)
**Sex** male (%) female (%)			
	6 (60%)	5 (50%)	6 (60%)
	4 (40%)	5 (50%)	4 (40%)
**Osteolytic bone** **lesions** yes (%) no (%) unknown (%)			
	6 (60%)	5 (50%)	8 (80%)
	3 (30%)	4 (40%)	2 (20%)
	1 (10%)	1 (10%)	-
**Cytogenetics** hyperdiploid (%) non-hyperdiploid (%) gain 1q (%) inconclusive (%) normal (%)			
	3 (30%)	3 (30%)	3 (30%)
	1 (10%)	3 (30%)	4 (40%)
	2 (20%)	3 (30%)	4 (40%)
	6 (60%)	4 (40%)	2 (20%)
	-	-	1 (10%)

**Table 2 cancers-14-04901-t002:** Identification of previously described genetic variants in HMCLs. The detectability of previously described genetic variants in HMCLs was used as a positive control for our targeted gene sequencing technique. For each variant, its genomic position together with gDNA and protein change are indicated. Abbreviations used: VAF = variant allele frequency.

Cell Line	Gene	Position	gDNA Change	p. Change	VAF	Reference
OPM-2	*DIS3*	chr13:73355008	g.73355008T>G	p.Tyr121Ser	100% of 328 reads	Leich et al. [39]
OPM-2	*FAM46C*	chr1:118166023	g.118166023A>C	p.Glu178Ala	100% of 135 reads	Zhu et al. [40]
U266	*BRAF*	chr7:140453132	g.140453132T>A	p.Lys601Asn	65.4% of 665 reads	Lionetti et al. [41]
U266	*TP53*	chr17:7578449	g.7578449C>T	p.Ala161Thr	100% of 344 reads	Moreaux et al. [42]
RPMI-8226	*KRAS*	chr12:25398284	g.25398284C>G	p.Gly12Ala	100% of 233 reads	Moreaux et al. [42]

**Table 3 cancers-14-04901-t003:** Somatic variants selectively detected in BM-DNA. This table gives an overview of the somatic variants that were only detected in BM-DNA (*n* = 6) and in neither of the matched liquid biopsy-derived DNA samples. NA = not available.

Patient	Gene	cDNA Change	p. Change	Classification	% PCs BM	% Monoclonal Ig seq BM
2019-008	*FAM46C*	c.823_825del	p.Phe275del	VUS	18%	49.1%
2019-015	*PIK3CA*	c.3125_3128del	p.Gln1042Argfs*25	VUS	4.2%	51.9%
2019-015	*KRAS*	c.183A>C	p.Gln61His	Pathogenic		
2020-024	*RB1*	c.610G>T	p.Glu204*	Likely pathogenic	7.1%	38.4%
2020-025	*CDK12*	c.1132C>T	p.Arg378Cys	VUS	3.8%	59.3%
2021-010	*KRAS*	c.183A>C	p.Gln61His	Pathogenic	NA	NA

**Table 4 cancers-14-04901-t004:** Somatic variants selectively detected in circulating biomarkers. This table gives an overview of the somatic variants that were only detected in circulating biomarker DNA (*n* = 14) and not in matched BM-DNA samples. Patients in whom extramedullary disease was strongly suspected are indicated with a dagger. NA = not available.

Patient	Gene	cDNA Change	p. Change	Classification	% PCs BM	% Monoclonal Ig seq BM
2018-001	*RAD50*	c.3751G>A	p.Glu1251Lys	VUS	NA	45.3%
2018-005 †	*NRAS*	c.182A>G	p.Gln61Arg	Pathogenic	NA	0.7%
2018-005 †	*FAM46C*	c.416_417dup	p.Gln140Cysfs*4	Likely pathogenic		
2018-005 †	*CBL*	c.2662G>A	p.Ala888Thr	VUS		
2018-011	*ATRX*	c.3145A>G	p.Ile1049Val	VUS	NA	50.5%
2018-017	*MTOR*	c.3286-1G>C	splice site	VUS	NA	60.3%
2019-007 †	*KRAS*	c.34G>T	p.Gly12Cys	Pathogenic	NA	NA
2019-007 †	*DIS3*	c.2339G>A	p.Arg780Lys	VUS		
2020-004	*STK11*	c.938A>G	p.His313Arg	VUS	NA	16.0%
2020-024	*NRAS*	c.183A>T	p.Gln61His	Pathogenic	7.1%	38.4%
2020-024	*NRAS*	c.38G>T	p.Gly13Val	Pathogenic		
2020-024	*NRAS*	c.38G>A	p.Gly13Asp	Pathogenic		
2020-024	*RB1*	c.2107-1G>A	splice site	Likely pathogenic		
2022-003	*TSC1*	c.2393C>T	p.Thr798Met	VUS	NA	NA

## Data Availability

The raw sequencing data generated in this study can be accessed at the European Nucleotide Archive under accession number PRJEB56136.

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
