# Peer review of "Liquid Biopsy-Derived DNA Sources as Tools for Comprehensive Mutation Profiling in Multiple Myeloma: A Comparative Study"

_cancers, 2022, doi:10.3390/cancers14194901_

Round 1
Reviewer 1 Report
In my opinion the manuscript is clearly written, the data are relevant to the field and the results sustain their conclusion.
I have no further specific recommendations or comments.
Author Response
- In my opinion the manuscript is clearly written, the data are relevant to the field and the results sustain their conclusion. I have no further specific recommendations or comments.
Response to the reviewer:
The authors thank the reviewer for the positive feedback regarding our submitted manuscript.
Reviewer 2 Report
Heestermans et al. compare the mutational profile of Multiple Myeloma (MM) patients derived from different sources of DNA from liquid biopsies (cfDNA, EV-DNA, CTC-DNA and PBMCs-DNA) with the profile from the bone marrow (BM-DNA). The genetic alterations were evaluated through a targeted panel of 165 genes, that included both somatic and germline mutations. The authors conclude that cfDNA is the best circulating biomarker for genetic characterization, and that a combined analysis of cfDNA and BM-DNA would allow for a complete profiling of MM. Interestingly, the DNA source from EVs was investigated for the first time for MM patients.
The importance of liquid biopsies in MM, lies on the fact that an invasive bone marrow biopsy could be avoided for the welfare of the patients. The results from this study highlights that this could probably be achieved through the study of cfDNA, as the best DNA source for genetic alterations, at least for the 165 genes evaluated.
Comments:
How was the panel of 165 genes decided? Was it previously used? Why include germline mutations in the panel? It is interesting but strange when you want to perform targeted sequencing of a small number of genes. Were these germline alterations already described before? It would be of interest if the information on why these 165 genes were chosen is described.
How are these genes correlated with prognosis and alterations included in R-ISS? This is of interest if this panel is thought to have the potential to be used in the clinics.
Why analyze all PBMCs? It would be expected that somatic mutations would be specific of the malignant PCs (CD138+ CTCs) and would be diluted in all cells, and therefore less likely to be detected.
In Discussion:
1st sentence – The first-full scale comparison – It gives the idea that it is whole-genome when it is targeted to 165 genes.
When comparing results with other published results, what type of sequencing did they use (panel)? Is it even comparable? If not, please clarify.
It is a bit confusing the discussion and information regarding the pathway analysis. The genes included in the panel were chosen because they somehow already correlate with MM? Then the pathways dysregulated are already known and biased for that.
Author Response
- How was the panel of 165 genes decided? Was it previously used? Why include germline mutations in the panel? It is interesting but strange when you want to perform targeted sequencing of a small number of genes. Were these germline alterations already described before? It would be of interest if the information on why these 165 genes were chosen is described.
Response to the reviewer:
These are important and valuable remarks made by the reviewer. The panel we used for targeted gene sequencing contains 165 genes that are of interest in solid and hematological malignancies. This panel was designed by BRIGHTcore (Vrije Universiteit Brussel (VUB), Université Libre de Bruxelles (ULB)) with whom we collaborate and is currently used in the diagnostic work-up of oncology patients within our institution UZ Brussel. At the time this panel was designed, different departments and oncology research teams within our institution could propose genes to be included in the 165 gene panel. Specifically for this study, we included genes that are frequently mutated in multiple myeloma (MM) and/or associated with therapy resistance based on a thorough review of the literature, including KRAS, NRAS, TP53, FAM46C, DIS3, BRAF, CRBN, CUL4B, IRF4, IKZF1, ATM, ATR, CCND1, EGR1, HIST1H1E, TRAF3, CYLD, RB1, FGFR3, DNMT3A and LTB. However, the exact composition of the 165 gene panel was not based on previous use of this panel in the literature but was designed to be applicable in several hematological and solid malignancies. This implies that the panel also contains genes that are of a minor interest in MM. However, the goal of this study was to investigate the applicability of liquid biopsy-derived DNA sources for mutation profiling in MM. As it was not our first intent to study the genetic alterations in MM on a genome-wide scale, we chose to use the clinically relevant and validated 165 gene panel for our experiments. An informative sentence regarding the composition of the 165 gene panel was added to line 147-148, page 4 of the revised version of the manuscript.
We based our filtering and interpretation of the raw sequencing data on the guidelines established by the ComPerMed working group as discussed in the manuscript. The filtering based on these guidelines do not exclude germline variants from reporting. Only germline and somatic variants that can be classified as benign or likely benign and/or that do not meet technical requirements for coverage and VAF are excluded from reporting based on these guidelines. The majority of the germline variants we reported have already been described in population databases and registries such as Gnomad, dbSNP and Clinvar, although not necessarily at germline VAF. As the biological significance of all germline variants we reported was assessed as VUS or likely pathogenic, the precise impact of most of these variants remains unclear. However, as a pathogenic effect of these variants cannot be excluded based on the classification we provided, we felt that we needed to include these variants in our reported results. The available number of studies focusing on and/or reporting germline variants in MM is still limited, as discussed in the manuscript. We hope that by reporting the observed germline variants, we can contribute to increase the knowledge regarding this subject.
- How are these genes correlated with prognosis and alterations included in R-ISS? This is of interest if this panel is thought to have the potential to be used in the clinics.
Response to the reviewer:
Thank you for this interesting remark. The R-ISS staging system currently describes the following cytogenetic alterations as high-risk and associated with inferior prognosis: t(4;14), t(14;16) and del(17p). It is well-established that TP53 inactivation by del(17p) is associated with inferior prognosis, but when TP53 mutations occur in combination with del(17p), this confers to an even worse prognosis. Chesi et al. (Blood, 1998) and Intini et al. (British Journal of Haematology, 2001) earlier described the occurrence of activating FGFR3 mutations in MM cell lines and patients harboring t(4;14). The inferior prognostic impact of somatic variants in ATM, ATR, CCND1 and TP53 and the positive impact of variants in IRF4 were previously described by Walker et al. (Journal of Clinical Oncology, 2015). Regardless of the gene that is mutated, Miller et al. (Blood Cancer Journal, 2017) demonstrated that patients with an above average amount of somatic variants had a significantly shorter PFS than those with a below average amount of somatic variants. Of note, Kortum et al. (Blood, 2016) described mutations in CRBN and genes of the CRBN pathway in a significant amount of refractory MM patients after receiving treatment with IMiD’s and related mutations in these genes to decreased sensitivity to IMiD’s. As all genes mentioned in this paragraph are prognostically relevant in MM and are included in the 165 gene panel we used, we believe it can be of additional value in the diagnostic work-up of multiple myeloma patients when applied in a clinical setting. We included data regarding the patient cytogenetic alterations in column Q of Supplementary Table 2. However, as FISH and/or CGH was not performed for a significant number of patients in our study population, we chose not to correlate the mutational load with the R-ISS.
- Why analyze all PBMCs? It would be expected that somatic mutations would be specific of the malignant PCs (CD138+ CTCs) and would be diluted in all cells, and therefore less likely to be detected.
Response to the reviewer:
The authors agree with the reviewer that somatic variants occurring in MM patients specifically affect the malignant PCs and are more likely to be detectable in MNCs enriched for CD138 positivity. However, as Coffey et al. (Blood Cancer Journal, 2019) used unenriched PBMNCs successfully to detect MM-associated somatic variants, we included both unenriched PBMNCs and CTC enriched for CD138 positivity in our analysis to be able to compare the performance of both. A shown in Figure 4 and discussed in lines 469-471, page 13 of the revised manuscript, our data clearly showed the added value of immunomagnetic enrichment of CTCs from PBMNCs to detect somatic variants.
- Discussion 1st sentence – The first-full scale comparison – It gives the idea that it is whole-genome when it is targeted to 165 genes.
Response to the reviewer:
The authors agree with the reviewer that this sentence can be confusing to the reader. It has been corrected on lines 417-419, page 12 of the revised manuscript.
- When comparing results with other published results, what type of sequencing did they use (panel)? Is it even comparable? If not, please clarify.
Response to the reviewer:
This is an interesting remark regarding the sequencing methodology we used. When reviewing the methods used by other sequencing studies cited in our manuscript, about fifty percent of them used whole-exome sequencing whereas the other half used targeted gene sequencing (TGS) as we did. Both TGS and whole-exome sequencing permit to evaluate and compare the performance of liquid biopsy-derived DNA sources. Restricting the analysis to a limited number of genes by using TGS does not alter the biological/genetic differences between the different DNA sources we observed, making our results comparable to studies using whole-exome sequencing or TGS. TGS has the advantage that it focusses on a limited number of genes, permitting to reduce costs and data load compared to whole-exome or whole-genome sequencing. As it focusses on genes that are known to be of pathological importance, it is probably the most efficient way to use low quantity DNA samples such as EV-DNA (and cfDNA) when applied in a clinical setting.
- It is a bit confusing the discussion and information regarding the pathway analysis. The genes included in the panel were chosen because they somehow already correlate with MM? Then the pathways dysregulated are already known and biased for that.
Response to the reviewer:
Thank you for this valuable remark. As was mentioned above, about 21 genes that are frequently involved in MM were indeed included in the 165 gene panel we used. This however includes genes like NRAS KRAS and TP53, which are not specific for MM and are often mutated in other types of malignancies. The gene panel also contains a large number of tumor-related genes that are more frequently involved in other types of solid and/or hematological malignancies and are less frequently affected in MM (see also Supplementary Table 1 for a complete gene list). As a result and as we discussed above, the gene panel was not designed to be completely myeloma-specific but to be applicable to a broad range of solid and hematological tumors. This implies that the panel itself is not enriched on purpose for genetic pathways that are most frequently affected in MM. NRAS, KRAS and TP53 were among the most frequently mutated genes in our study, as was also observed in other studies in MM patients. This results in the PI3K/Akt, MAPK and p53 signaling pathways being affected by a high number of somatic variants, as these pathways are related to the genes mentioned above. However, these pathways have previously been identified as some of the most commonly affected pathways in MM in independent sequencing studies. We hope this explanation provides the necessary clarifications. We chose to address only briefly the results regarding the gene pathways analysis but if requested, this elaboration can be added to the discussion.
Reviewer 3 Report
Dear Authors,
This is a very timely study of the various sources for liquid biopsy samples for mutational profiling for multiple myeloma patients, and analysis of the mutations identified. The findings provide valuable information from both a technical perspective, as well as insight into mutational load with disease progression and repeated treatments.
Below are some points that need to be addressed to improve the manuscript.
-Please summarize the sequencing depth achieved with each sample type (provide mean/median and range of coverage).
-Line 196: You indicate that variants with MAF of >0.1% were excluded. Did you mean to say MAF of < 0.1% were excluded?
-Please include ethnicity/race information for each patient in the Supplemental Table 2.
-Please include information on the treatments used in the patients as first, second and third line of treatments, so the additional mutational burden in the relapsing patients can be better evaluated in this and in future studies. If there are cases with different treatments in this study, discuss the potential impact of the treatments on the mutational load.
-In Supplementary Table 3 and Supplementary Table 4, please review the use of the “coma” to indicate a decimal point instead of the “period” to designate a decimal point. At the moment, it appears some cells have a coma and others a period. This can be confusing.
Author Response
- Please summarize the sequencing depth achieved with each sample type (provide mean/median and range of coverage).
Response to the reviewer:
We agree with the reviewer that information regarding the sequencing depth of the different sample types is of importance for the reader. Although the mean target coverage was already mentioned in the manuscript, we added the range of mean target coverage for each sample type on lines 329-331, page 9 of the revised manuscript.
- Line 196: You indicate that variants with MAF of >0.1% were excluded. Did you mean to say MAF of < 0.1% were excluded?
Response to the reviewer:
Thank you for this comment. We indeed excluded variants with a population MAF (minor allele frequency) >0.1% according to the recommendations of the ComPerMed guideline. This guideline considers variants with a MAF >0.1% to be common variants in the population which results in their classification as “benign” or “likely benign”. They are excluded from reporting when applied in a clinical setting and as such we also did not include them in our data analysis.
- Please include ethnicity/race information for each patient in the Supplemental Table 2
Response to the reviewer:
This is a valuable remark made by the reviewer. The information regarding the ethnicity of the patients included in our study was added to column D in the revised version of Supplementary Table 2.
- Please include information on the treatments used in the patients as first, second and third line of treatments, so the additional mutational burden in the relapsing patients can be better evaluated in this and in future studies. If there are cases with different treatments in this study, discuss the potential impact of the treatments on the mutational load.
Response to the reviewer:
The treatments received by the patients included in our study in first, second and third line are added to column I in the revised version of Supplementary Table 2 as requested by the reviewer. As can be derived from the data added to Supplementary Table 2, the treatments received differed between virtually all patients. As such, identification of differential mutational loads and patterns associated with different treatment regimes is very hard because of the limited number of patients receiving a specific combination of treatments. As the number of patients with second or later relapse is limited to ten in our study, the authors agree with the reviewer that a study investigating the impact of different treatment regimes on the mutational burden in relapsing MM patients would be highly interesting. However, given the limitations discussed above and the short amount of time available to revise the manuscript, it was not possible for us to perform such an in-depth analysis.
When entering the data about the treatment lines in Supplementary Table 2, we noticed that for patient 2022-010 (first relapse) the total number of treatment lines was listed as 2 in the table. However, this patient only received one treatment line prior to sample collection, which we corrected in column I in the revised version of Supplementary Table 2. The second treatment line was started after the sample collection took place. We recalculated the Spearman rank correlation coefficient between the number of somatic variants and the number of treatment lines received but the results remained unchanged, so no correction is needed in the manuscript text where these results are described (lines 314-316, page 8 of the revised manuscript).
- In Supplementary Table 3 and Supplementary Table 4, please review the use of the “coma” to indicate a decimal point instead of the “period” to designate a decimal point. At the moment, it appears some cells have a coma and others a period. This can be confusing.
Response to the reviewer:
Thank you for this valuable remark. As we used a “period” to indicate a decimal point in the manuscript text, we corrected this where necessary in all cells in the revised version of Supplementary Table 3 and 4.